# Copper and Zinc Content in Infant Milk Formulae Available on the Polish Market and Contribution to Dietary Intake

**DOI:** 10.3390/nu13082542

**Published:** 2021-07-25

**Authors:** Małgorzata Dobrzyńska, Sławomira Drzymała-Czyż, Karol Jakubowski, Szymon Kurek, Jarosław Walkowiak, Juliusz Przysławski

**Affiliations:** 1Department of Bromatology, Poznan University of Medical Sciences, 42 Marcelińska Street, 60-354 Poznań, Poland; drzymala@ump.edu.pl (S.D.-C.); jakubowski.karol@ump.edu.pl (K.J.); jprzysla@ump.edu.pl (J.P.); 2Department of Pediatric Gastroenterology and Metabolic Diseases, Poznan University of Medical Sciences, 27/33 Szpitalna Street, 60-572 Poznań, Poland; skurek@ump.edu.pl (S.K.); jarwalk@ump.edu.pl (J.W.)

**Keywords:** micronutrients, infant formula, follow-on formula, formula for special medical purposes, daily intake

## Abstract

The inappropriate concentration of copper (Cu) and zinc (Zn) in formulae for infants can lead to abnormal micronutrient intake and adverse health outcomes. This study aimed to determine the concentration of Cu and Zn in different formulae and evaluate the Cu/Zn ratio. Besides, the daily intake (DI) of both micronutrients was estimated. Cu and Zn concentration in 103 formulae for infants, available in the Polish market, were assessed using atomic absorption spectrometry. The estimated DI was calculated from the average energy requirements for the 0–6 months aged infants. The microelement content of formulae was mostly in good agreement with that declared by the manufacturer (5–10% variations compared to the labeled values). The Cu/Zn ratio ranged from 1:8 to 1:25. The estimated DI of Cu was in the range of 0.14–1.11 mg/day. Six (6.7%) of the formulae did not meet the recommended range of Cu intake, especially during the first month of life and in the case of formulae for special medical purposes. The estimated DI of Zn varied from 2.27–11.25 mg/day. In most cases, the concentration of Cu and Zn in infant formulae was within the recommended range. It would be advisable to consider monitoring the DI of Cu and reconsider the Cu content in formulae for infants in proportion to its expected consumption.

## 1. Introduction

Following the international nutrition guidelines for infants, exclusive breast-feeding for the first six months of life is recommended [1]. Breast milk provides an adequate supply of nutrients to support growth, development, and health. However, due to some constraints, breastfeeding is not possible [2,3]. In this situation, breast milk substitutes are needed. Nowadays, there are various complete formulae for infants available on the market. According to the European Union Directives, we can distinguish several types of formulae for infants; infant formula, follow-on formula, and formula for special medical purposes intended for infants. 

Infant formulae are used by infants during the first months of life (generally from 0–6 months) and satisfy nutritional requirements until the introduction of complementary foods. Follow-on formulae are used by infants after the introduction of complementary feeding (generally from 7–12 months) and constitute the principal liquid element in a diversified diet. The formulae for special medical purposes (from birth) are used by infants with special medical conditions, such as special disorders or diseases. Depending on the type of formula, the nutrient content varies. However, all formulae intended for infants must be safe and meet nutritional requirements [4]. The Scientific Committee on Food (SCF), European Food Safety Authority (EFSA), and European Union Directive and Regulation (Directive 2006/141/EC, Regulation 2016/128) have established minimum and maximum values for ingredients in formulae for infants [4,5,6]. A concentration of elements in formulae for infants, which are either too low or too high, can lead to adverse health outcomes (such as impaired growth and cognitive development, increased morbidity, and hyperactivity) [7].

Cu and Zn are essential nutrients for infants’ health. Excessive Zn intake can reduce intestinal copper absorption [8]. Furthermore, Zn in formulae can be less available than in the case of breast milk [9]. Zn plays a significant role in the regulation of cell division, cellular immunity, and sexual maturation. Zn deficiency symptoms include impaired growth and altered cognition in children, diarrhea, loss of appetite, susceptibility to infections, and skin lesions [10]. Excessive Zn is generally considered to be relatively nontoxic [11]. Cu is required for cellular metabolism in enzymatic and non-enzymatic systems [12]. In infants, Cu deficiency is rare, but has been reported in preterm infants and in infants recovering from malnutrition accompanied by diarrhea. A decrease of Cu causes impairment of growth, neutropenia, anemia, and increased risk of infection [13]. Toxic effects associated with increased chronic exposure to Cu comprise acute gastrointestinal symptoms, such as abdominal pain, vomiting, and diarrhea [14,15].

The main objectives of this study were to analyze Cu and Zn concentration in formulae for infants available in the Polish market and evaluate the Cu/Zn ratio. The daily intake (DI) of both micronutrients was also estimated and the reliability of the manufacturer’s declaration and repeatability in different batches were evaluated.

## 2. Materials and Methods

### 2.1. Materials

This study covered all available powdered milk formulae samples in the Polish market in the years 2019–2020. A total of 90 different formulae, designed for consumption during the first year of life, were analyzed, including 30 infant formulae, 29 follow-on formulae, and 31 formulae for special medical purposes for infants. In addition, 13 selected formula samples that came from a different batch in the following year were included in the study. Hence a total of 103 samples were analyzed. This study analyzed complete breast milk substitutes for special medical purposes, i.e., extensively hydrolyzed, amino acid-based, preterm, anti-reflux, lactose-free, medium-chain triglycerides, and phenylalanine-free formulae. Samples were coded and stored before analysis in conditions complying with the requirements of the manufacturers. Certified reference material—skimmed milk powder ERM—BD150 (European Reference Materials, Belgium) was used to validate the analytical methods.

### 2.2. Sample Preparation and Determination of Elements

The concentration of Cu and Zn in 103 formula samples for infants was determined using an atomic absorption spectrometer. The determination was based on the method described by Yaman et al. [16]. Sample digestion was carried out using a microwave accelerated reaction system (MARS 6, CEM Corporation, Matthews, NC, USA). Each sample (1 g) was weighed directly into a microwave vessel and filled with 10 mL 69% ultra-pure nitric acid (ROMIL, Cambridge, UK). Vessels were gently mixed and left open for 15 min to allow sample pre-digestion. The microwave heating program consisted of 2 steps; ramping from ambient temperature to 180 °C over 20 min, and then holding the temperature for 20 additional minutes. After cooling, the obtained solutions were transferred to 50 mL volumetric flasks and diluted to 20mL with ultrapure water. The atomic absorption spectrometry method (iCE 3000 Series, AAS, Thermo Scientific, Cambridge, UK) was used to determine the concentration of both Cu and Zn in samples. All analyses were performed in triplicate. The methods were validated by a simultaneous analysis of the reference material, with accuracy for Cu of 94.5% and for Zn of 95.4%.

### 2.3. Data Analyzes

The concentration of Cu and Zn (mg/100 g powder) in various types of formulae was compared to the declaration. The nutrients’ values declared on the label were mainly expressed in 100 mg of powder. On the labels, where the values were expressed per 100 mL for ready-to-use formula, the data were converted using the information on the label.

Amounts of Cu and Zn were converted to 100 kcal formulae. They were compared to the SCF, EFSA, and European Directive guidelines regarding minimum and maximum element content in formulae for infants.

To estimate the daily required intake (DRI) of Cu and Zn, the EFSA recommendations regarding energy intake values for infants were used [4]. Results were compared to nutritional guidelines for infants. The DRI of Cu and Zn were calculated only for infant formulae and formulae for special medical purposes intended for infants from birth. The estimated daily intake was calculated from the average energy requirements for the 0–6 months aged infants, declared by the EFSA NDA Panel, using the information on the labels on energy content and amount of powder needed to prepare 100 mL formula [4].

For all analyzed types of formula, the median, first quartile, third quartile, minimum and maximum values were calculated. Statistical analyses were performed using software Statistica 13.3 (StatSoft, TIBCO Software Inc., Palo Alto, CA, USA).

## 3. Results

The content of these elements in mg/100 g powder in each infant formula and producer declaration is presented in the Appendix A. The elemental content in analyzed formulae was mostly in good agreement with the declared content given by the manufacturer. The levels of Cu and Zn found in our study showed 5–10% (±) variation compared to the labeled values. Cu and Zn concentration by formula type is presented in Table 1. The concentration of Cu and Zn in the analyzed formulae varied from 0.20–1.00 mg/100 g powder and from 3.22–10.10 mg/100 g powder, respectively. The Cu/Zn ratio in formulae samples ranged between 1:8 and 1:25. 

Differences in Cu and Zn concentrations between two batches in thirteen formulae are illustrated in Figure 1 and Figure 2. In most of the analyzed samples, the concentration of Cu and Zn in different batches were satisfactory (19% difference was adopted) [17], median (1st–3rd quartile), Cu—0.33 (0.29–0.39) and Zn—4.13 (3.83–4.97); however, in one formula the Cu concentration difference was 33%.

Measured concentration of Cu and Zn (mg/100 kcal) is reported in Table 2. The nutritional requirements for minimum and maximum Cu and Zn content in formulae for infants are presented in Table 3 [4,5,6]. While comparing these limits with our results in two analyzed formulae, we found Cu exceeding the recommended limit in preterm formulae and phenylalanine-free formulae. All analyzed infant formulae and follow-on formulae met the minimum Cu level. However, in four formulae for special medical purposes, Cu concentration was below the limit: in one anti-reflux formula, two comfort formulae, and in one extensive hydrolyzed formula. The Zn concentration in all formulae met the recommended minimum and maximum levels.

According to the nutritional requirements, the adequate intake (AI) of Cu is 0.20 mg/day for infants of 0 to 6 months [18]. The estimated DI of Cu for infants (0–6 months) varied from 0.14–1.11 mg/day (Table 4). The estimated DI for some formulae was lower than the AI (formula with a minimal amount of Cu—0.14 mg/day), especially during the first month of life. On the other hand, in the case of two formulae for special medical purposes, the estimated daily intake of Cu exceeded upper level (UL) recommended for 1 year old infants [19] in the 5th month of life. 

The AI of Zn is 2 mg/day for infants of 0 to 6 months [18]. The estimated DI of Zn varied from 2.27–11.25 mg/day.

## 4. Discussion 

Over the years, the composition of infant milk formula has been constantly changing, which suggests the need to repeat previous analyzes [9,20,21]. This is the largest study to assess Cu and Zn content in all powdered milk formulae for infants available on the Polish market. In most cases, the concentration of micronutrients in infant formulae was within the recommended range. Problems with breastfeeding make milk formulae a main, or only, source of nutrition for infants. Currently, there is a wide range of types and brands of formula for infants available in the market. According to FDA (Food and Drug Administration), all formulae are required to satisfy the quality factors for normal growth and health [22]. The content of all nutritional ingredients in formulae is regulated and monitored to meet international quality criteria [23]. 

Cu and Zn are microelements which take part in many metabolic processes. The Cu and Zn content of breast milk is highest during early lactation and then declines during the course of lactation [24]. If formulae are used, the content of these elements will be higher. Zn in formulae can be less bioavailable than in breast milk. Moreover, absorption of these ingredients is interdependent.

All available powdered formulae for infants in the Polish market were analyzed in this study. The content of Cu and Zn in the samples varied. No significant differences were observed between the values measured and those declared on the labels. The concentration of Cu and Zn were mostly within the recommended range. Nevertheless, one preterm formula and one phenylalanine-free formula exceeded the permissible limit. In the preterm formula, the Cu content was exceeded by 15% of the maximum limit, and the estimated DI from 2–6 months was three times higher than the adequate intake. Zn content in this formula was in line with the requirement, but the estimated DI of Zn in the diet in the fifth and sixth months was too high (higher than the UL for 1 year old children—1 mg/day). However, it is worth noting, preterm infants have concomitant increased requirements for Cu and Zn [25,26]. In the phenylalanine-free formula, the Cu content was exceeded by 65% of the maximum limit, and the estimated DI was high (in the fifth and sixth month were higher than the UL for 1 year old children—7 mg/day). The Zn content in this formula was in line with the requirement, but the estimated supply of Zn in the diet within the first month of life was higher than 9 mg/day. The minimum Cu level was not met for some formulae for special medical purposes: one anti-reflux formula, two comfort formulae, and one extensive hydrolyzed formula. This is probably related to the differences between the recommended minimum levels of copper in formulae for special medical purposes for infants (0.06 mg/100 kcal) and “regular” infant formulae (0.035 mg/kcal) and follow-on formulae (0.035 mg/kcal). All the four analyzed formulae for special medical purposes met the limits for copper in infant and follow-on formulae. The underlying reason might be that some of the formulae for special medical purposes for infants are intended to be used both by infants aged 0–6 and 6–12 months as declared by the manufacturer.

The estimated DI of Cu for infants (0–6 months) varied. In most of the analyzed formulae, the estimated DI of Cu in the first month was similar to the AI and increased proportionally in the following months. The estimated DI of Zn for most formulae from two months was twice the AI. The estimated DI of Zn was significantly higher for preterm, anti-reflux, and amino acid-based formulae, and the estimated DI of Cu was significantly higher for preterm and amino acid-based formula. This may be related to the lower bioavailability of zinc from formula for infants. Zinc that is bound to casein is less bioavailable compared with zinc bound to whey proteins [27].

The large variation in the Cu/Zn ratio is also worth noting. Recommendations for the Cu/Zn ratio in infant formulae are not specified. However, in breast milk, a Cu/Zn ratio changes at different stages of lactation. It is highest in colostrum (1:8); it decreases to 1:5 in the first month of lactation and 1:3.5 in the sixth month of lactation. [9,28]. The August et al. study showed that when Cu to Zn ratios of 1:2, 1:5, and 1:15 were fed to humans, there were limited effects on Cu absorption [29]. Ehrenkranz et al. reported that a Cu/Zn ratio of 1:20 (0.4 mg/L Cu; 8.0 mg/L Zn) produced a significantly higher excretion of Cu [30]. It has not been reported that a high Cu/Zn ratio can cause adverse health effects [31]. Some studies suggest that the serum Cu/Zn ratio may affect cognitive functioning in early life [32].

While analyzing the content of microelements in formula for infants (powdered), quality and content of Cu in drinking water should also be considered. Cu concentration in drinking water varies widely [33]. Both bottled water and drinking water can affect the Cu content of the ready-mix milk formula. The WHO guideline threshold limits for Cu for bottled water and drinking water are 1.0 mg/L and 2.0 mg/L, respectively [34].

Since serum microelement concentrations were not measured, we could not assess Zn and Cu absorption. In our study, we analyzed 90 samples from all formulae available on the Polish market; in the cases of 13 formulae the measurement was repeated on samples derived from separate batches. Unfortunately, due to the lack of sufficient funds, we were not able to analyze more samples from different batches of the same formula. On the other hand, analyzing all formulae available on the market, and on this basis estimating the daily intake of Cu and Zn, could have an impact on the nutritional recommendations.

## 5. Conclusions

In most cases, the concentration of Cu and Zn in the analyzed formulae for infants was within the recommended range and the manufacturers provided accurate declarations of values. Preparations for special medical purposes have higher requirements, which were not always met. It would be advisable to consider monitoring the DI of Cu and reconsider the Cu content in the formulae for infants to fulfil its expected consumption.

## Figures and Tables

**Figure 1 nutrients-13-02542-f001:**
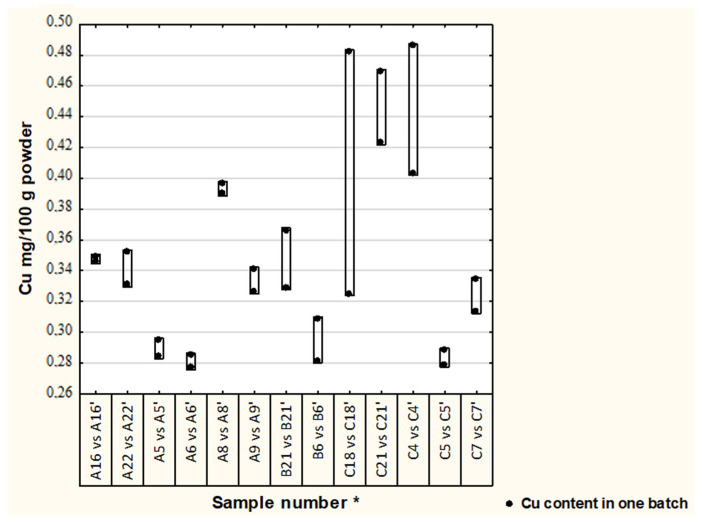
Differences in Cu concentration between batches of analyzed formulae. Second batch of the same formula, * randomly assigned formulae symbols from two other batches

**Figure 2 nutrients-13-02542-f002:**
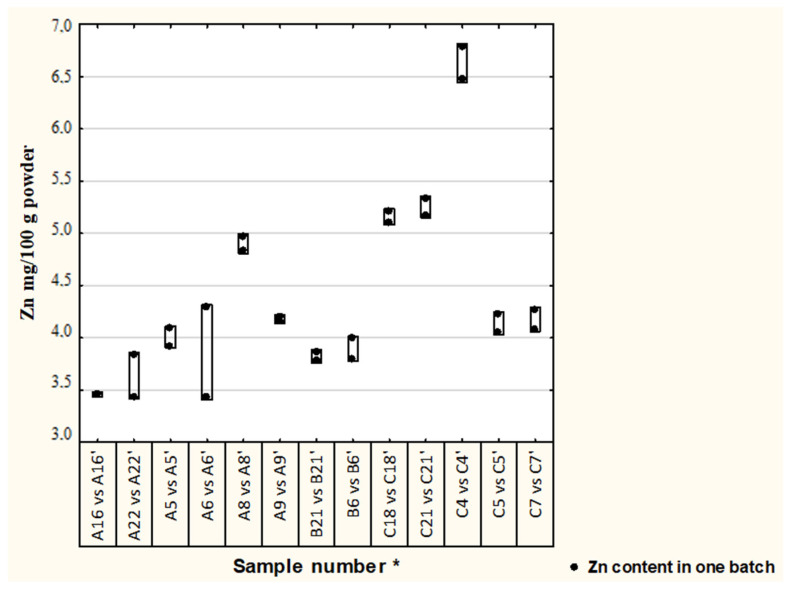
Differences in Zn concentration between batches of analyzed formulae. Second batch of the same formula, * randomly assigned formulae symbols from two other batches

**Table 1 nutrients-13-02542-t001:** Copper and Zinc concentration (mg/100 g powder) in different types of formula for infants.

Type of Formulae for Infants	Number of Samples (Number of Different Formulae) *	Cu mg/100 g Powder	Zn mg/100 g Powder	Cu/Zn
		Median (1st–3rd Quartile) <Min–Max>		
Total formulae for infants (0–12 months)	103 (90)	0.34 (0.29–0.37)	4.18 (3.83–5.09)	1:13 (1:11–1:15)
		<0.20–1.00>	<3.22–10.10>	<1:8–1:25>
Infant formulae (0–6 months)	36 (30)	0.33 (0.29–0.36)	4.00 (3.64–4.29)	1:12 (1:11–1:14)
		<0.23–0.43>	<3.31–6.11>	<1:9–1:19>
Infant formulae manufactured from cows’ milk proteins	25 (21)	0.34 (0.30–0.36)	3.89 (3.64–4.22)	1:12 (1:11–1:14)
		<0.23–0.43>	<3.31–6.11>	<1:9–1:19>
Infant formulae manufactured from goats’ milk proteins	4 (3)	0.34 (0.27–0.39)	4.02 (3.58–4.90)	1:12 (1:12–1:13)
		<0.27–0.39>	<3.58–4.90>	<1:12–1:13>
Infant formulae manufactured from protein hydrolysates	7 (6)	0.33 (0.29–0.34)	4.13 (3.99–4.37)	1:13 (1:12–1:14)
		<0.28–0.36>	<3.36–4.72>	<1:10–1:15>
Follow–on formulae (7–12 months)	31 (29)	0.34 (0.28–0.36)	4.13 (3.83–4.59)	1:13 (1:11–1:15)
		<0.22–0.41>	<3.22–5.74>	<1:8–1:24>
Follow–on formulae manufactured from cows’ milk proteins	23 (21)	0.34 (0.29–0.36)	4.12 (3.83–4.32)	1:12 (1:11–1:15)
		<0.22–0.40>	<3.22–5.44>	<1:8–1:22>
Follow–on formulae manufactured from goats’ milk proteins	4	0.36 (0.31–0.39)	4.64 (4.46–4.73)	1:13 (1:12–1:15)
		<0.26–0.41>	<4.34–4.77>	<1:11–1:18>
Follow–on formulae manufactured from protein hydrolysates	4	0.27 (0.25–0.30)	3.88 (0.77–4.84)	1:14 (1:13–1:19)
		<0.24–0.32>	<3.70–5.74>	<1:12–1:24>
Formula for special medical purposes intended for infants	36 (31)	0.36 (0.32–0.42)	5.16 (3.89–5.72)	1:13 (1:12–1:16)
		<0.20–1.00>	<3.23–10.10>	<1:8–1:25>
Formula for special medical purposes intended for infants from birth	31 (26)	0.38 (0.32–0.41)	4.90 (3.80–5.20)	1:13 (1:11–1:15)
		<0.27–0.95>	<3.50–8.60>	<1:8–1:25>
Formula for special medical purposes intended for infants from 7 months	5	0.36 (0.28–0.37)	4.70 (3.50–4.90)	1:15 (1:12–1:16)
		<0.28–0.37>	<3.40–5.30>	<1:12–1:16>
Extensively hydrolyzed formulae	7 (6)	0.35 (0.29–0.36)	4.12 (3.87–5.37)	1:13 (1:11–1:16)
		<0.28–0.46>	<3.61–5.64>	<1:9–1:16>
Amino acid–based formulae	4 (2)	0.42 (0.40–0.45)	5.21 (5.16–5.25)	1:12 (1:12–1:13)
		<0.40–0.45>	<5.16–5.25>	<1:12–1:13>
Anti–reflux formulae	5	0.34 (0.33–0.35)	5.04 (4.96–5.16)	1:15 (1:14–1:16)
		<0.20–0.40>	<4.84–5.44>	<1:12–1:25>
Comfort–formulae	7	0.36 (0.27–0.42)	4.86 (3.74–5.37)	1:13 (1:12–1:14)
		<0.26–0.43>	<3.23–5.72>	<1:9–1:17>
Lactose–free formulae	2	0.40 <0.36–0.45>	4.92 <3.88–5.95>	1:12 <1:11–1:13>
Preterm formulae	4 (3)	0.44 (0.42–0.69)	6.63 (5.79–6.80)	1:15 (1:8–1:16)
		<0.42–0.69>	<5.79–6.80>	<1:8–1:16>
Soya–based formula for infants	1	0.30	5.75	1:19
Medium chain triglycerides formulae	3 (2)	0.30	3.94	1:13
		<0.28–0.32>	<3.69–4.18>	<1:13–1:13>
Phenylalanine–free formulae (complete)	3	0.37	6.56	1:11
		<0.35–1.00>	<3.72–10.10>	<1:10–1:18>

* For the same formula from a different batch, the average value was used.

**Table 2 nutrients-13-02542-t002:** Copper and Zinc concentration (mg/100 kcal formula) in different types of formula for infants (values that do not meet the recommendation are shown in bold).

Type of Formula for Infants	Number of Samples (Number of Different Formulae) *	Cu mg/100 Kcal Formula	Zn mg/100 Kcal Formula
		Median (1st–3rd Quartile) <Min–Max>	
Total formulae for infants (0–12 months)	103 (90)	0.069 (0.061–0.078)	0.86 (0.79–1.04)
		<0.040–**0.198**>	<0.66–2.01>
Infant formulae (0–6 months)	36 (30)	0.067 (0.059–0.072)	0.81 (0.73–0.92)
		<0.044–0.086>	<0.66–1.34>
Infant formulae manufactured from cows’ milk proteins	25 (21)	0.068 (0.059–0.072)	0.79 (0.72–0.99)
		<0.044–0.086>	<0.66–1.34>
Infant formulae manufactured from goats’ milk proteins	4 (3)	0.067 (0.055–0.077)	0.80 (0.73–0.96)
		<0.055–0.077>	<0.73–0.96>
Infant formulae manufactured from protein hydrolysates	7 (6)	0.066 (0.061–0.070)	0.85 (0.84–0.89)
		<0.057–0.072>	<0.66–0.92>
Follow–on formulae (7–12 months)	31 (29)	0.067 (0.059–0.080)	0.87 (0.82–0.97)
		<0.045–0.086>	<0.70–1.17>
Follow–on formulae manufactured from cows’ milk proteins	23 (21)	0.068 (0.062–0.081)	0.84 (0.82–1.00)
		<0.045–0.086>	<0.70–1.16>
Follow–on formulae manufactured from goats’ milk proteins	4	0.072 (0.061–0.077)	0.92 (0.89–0.95)
		<0.053–0.080>	<0.87–0.97>
Follow–on formulae manufactured from protein hydrolysates	4	0.059 (0.053–0.063)	0.85 (0.82–1.01)
		<0.050–0.064>	<0.80–1.17>
Formula for special medical purposes intended for infants	36 (31)	0.071 (0.065–0.083)	1.04 (0.81–1.14)
		<0.040–0.198>	<0.70–2.01>
Formula for special medical purposes intended for infants from birth	31 (26)	0.074 (0.065–0.085)	1.03 (0.82–1.14)
		<**0.040**–**0.198**>	<0.71–2.01>
Formula for special medical purposes intended for infants from 7 months	5	0.071 (0.062–0.073)	1.05 (0.76–1.14)
		<**0.056**–0.078>	<0.70–1.22>
Extensively hydrolyzed formulae	7 (6)	0.071 (0.062–0.078)	0.82 (0.77–1.14)
		<**0.057**–0.090>	<0.76–1.22>
Amino acid–based formulae	4 (2)	0.086 (0.083–0.089)	1.05 (1.04–1.06)
		<0.083–0.089>	<1.04–1.06>
Anti–reflux formulae	5	0.070 (0.065–0.071)	1.02 (1.01–1.05)
		<**0.040**–0.080>	<0.99–1.06>
Comfort–formulae	7	0.070 (0.057–0.081)	0.95 (0.78–1.11)
		<**0.056**–0.090>	<0.70–1.14>
Lactose–free formulae	2	0.078 <0.070–0.085>	0.95 <0.77–1.13>
Preterm formulae	4 (3)	0.091 (0.083–0.138)	1.35 (1.15–1.35)
		<0.083–**0.138**>	<1.15–1.35>
Soya–based formula for infants	1	0.060	1.16
Medium chain triglycerides formulae	3 (2)	0.064	0.83
		<0.063–0.065>	<0.82–0.85>
Phenylalanine–free formulae (complete)	3	0.080	1.41
		<0.067–**0.198**>	<0.71–2.01>

* For the same formula from a different batch, the average value was used.

**Table 3 nutrients-13-02542-t003:** Minimum and maximum Cu and Zn content requirement in formulae for infants per 100 kcal formulae [5,6].

Type of Formula	Cu (mg)		Zn (mg)	
	Per 100 Kcal Formula		Per 100 Kcal Formula	
	Minimum	Maximum	Minimum	Maximum
Infant formulae	0.035	0.1	0.5	1.5
Follow-on formulae	0.035	0.1	0.5	1.5
Formulae for special medical purposes for infants	0.060	0.12	0.5	2.4

**Table 4 nutrients-13-02542-t004:** Estimated daily intake (mg/day) of copper and zinc with different types of formula for infants (values that does not meet the recommendation are shown in bold).

Type of Formulae for Infants	Number of Samples (Number of Different Formulae) *	0 to <1 Month	1 to <2 Months	2 to <3 Months	3 to <4 Months	4 to <5 Months	5 to <6 Months
		Median (1st–3rd Quartile) <Min–Max>					
Total formulae for infants (0–12 months)	103 (90)						
Cu		0.24 (0.21–0.27)	0.33 (0.29–0.37)	0.35 (0.31–0.39)	0.33 (0.29–0.37)	0.36 (0.32–0.41)	0.39 (0.34–0.44)
		<**0.14**–0.68>	<**0.19**–0.95>	<0.20–0.99>	<**0.19**–0.95>	<0.21–**1.04**>	<0.23–**1.11**>
Zn		2.95 (2.73–3.59)	4.09 (3.79–4.97)	4.31 (3.99–5.24)	4.11 (3.81–5.00)	4.54 (4.17–5.48)	4.82 (4.46–5.85)
		<2.27–6.90>	<3.15–**9.57**>	<3.31–**10.07**>	<3.16–**9.61**>	<3.47–**10.53**>	<3.70–**11.25**>
Infant formulae (0–6 months)	36 (30)						
Cu		0.23 (0.20–0.25)	0.32 (0.28–0.34)	0.34 (0.30–0.36)	0.32 (0.28–0.34)	0.35 (0.31–0.38)	0.38 (0.33–0.40)
		<**0.15**–0.29>	<0.21–0.41>	<0.22–0.43>	<0.21–0.41>	<0.23–0.45>	<0.25–0.48>
Zn		2.80 (2.52–3.18)	3.88 (3.49–4.40)	4.08 (3.67–4.64)	3.89 (3.51–4.42)	4.27 (3.84–4.85)	4.56 (4.11–5.18)
		<2.27–4.62>	<3.15–6.41>	<3.31–6.74>	<3.16–6.43>	<3.47–**7.05**>	<3.70–**7.53**>
Infant formulae manufactured from cows’ milk proteins	25 (21)						
Cu		0.23 (0.20–0.25)	0.32 (0.28–0.34)	0.34 (0.30–0.36)	0.32 (0.28–0.34)	0.35 (0.31–0.38)	0.38 (0.33–0.40)
		<0.15–0.29>	<0.21–0.41>	<0.22–0.43>	<0.21–0.41>	<0.23–0.45>	<0.25–0.48>
Zn		2.73 (2.48–3.39)	3.79 (3.45–4.70)	3.99 (3.63–4.95)	3.81 (3.46–4.72)	4.17 (3.79–5.17)	4.46 (4.05–5.53)
		<2.27–4.62>	<3.15–6.41>	<3.31–6.74>	<3.16–6.43>	<3.47–**7.05**>	<3.70–**7.53**>
Infant formulae manufactured from goats’ milk proteins	4 (3)						
Cu		0.23 (0.19–0.26)	0.32 (0.260.37)	0.34 (0.27–0.39)	0.32 (0.26–0.37)	0.35 (0.29–0.40)	0.38 (0.31–0.43)
		<**0.19**–0.26>	<0.26–0.37>	<0.27–0.39>	<0.26–0.37>	<0.29–0.40>	<0.31–0.43>
Zn		2.77 (2.53–3.30)	3.84 (3.49–4.57)	4.04 (3.67–4.81)	3.85 (3.51–4.59)	4.22 (3.84–5.03)	4.51 (4.11–5.38)
		<2.52–3.30>	<3.49–4.57>	<3.67–4.81>	<3.51–4.59>	<3.84–5.03>	<4.11–5.38>
Infant formulae manufactured from protein hydrolysates	7 (6)						
Cu		0.23 (0.21–0.24)	0.32 (0.29–0.33)	0.33 (0.30–0.35)	0.32 (0.29–0.34)	0.35 (0.32–0.37)	0.37 (0.34–0.39)
		<0.20–0.25>	<0.27–0.34>	<0.29–0.36>	<0.28–0.34>	<0.30–0.38>	<0.32–0.40>
Zn		2.93 (2.87–3.05)	4.07 (3.98–4.23)	4.28 (4.19–4.45)	4.08 (4.00–4.25)	4.47 (4.38–4.66)	4.78 (4.69–4.97)
		<2.28–3.18>	<3.16–4.40>	<3.32–4.64>	<3.17–4.42>	<3.47–4.85>	<3.71–5.18>
Formula for special medical purposes intended for infants (from birth)	31 (26)						
Cu		0.25 (0.22–0.29)	0.35 (0.31–0.40)	0.37 (0.33–0.43)	0.35 (0.31–0.41)	0.39 (0.34–0.44)	0.42 (0.370.48)
		<**0.14**–0.68>	<**0.19**–0.95>	<0.20–0.99>	<**0.19**–0.95>	<0.21–**1.04**>	<0.23–**1.11**>
Zn		3.55 (2.81–3.93)	4.92 (3.89–5.45)	5.18 (4.10–5.74)	4.94 (3.91–5.47)	5.42 (4.29–6.00)	5.79 (4.58–6.41)
		<2.45–6.90>	<3.39–**9.57**>	<3.57–**10.07**>	<3.40–**9.61**>	<3.73–**10.53**>	<3.99–**11.25**>
Extensively hydrolyzed formulae	3						
Cu		0.24	0.33	0.35	0.33	0.36	0.39
		<0.20–0.31>	<0.27–0.43>	<0.29–0.45>	<0.27–0.43>	<0.30–0.47>	<0.32–0.51>
Zn		2.80	3.88	4.08	3.89	4.27	4.56
		<2.65–2.87>	<3.67–3.98>	<3.86–4.18>	<3.68–3.99>	<4.04–4.38>	<4.31–4.68>
Amino acid–based formulae	4 (2)						
Cu		0.30 (0.29–0.30)	0.41 (0.40–0.42)	0.43 (0.42–0.44)	0.41 (0.40–0.42)	0.45 (0.44–0.47)	0.48 (0.47–0.50)
		<0.29–0.30>	<0.40–0.42>	<0.42–0.44>	<0.40–0.42>	<0.44–0.47>	<0.47–0.50>
Zn		3.62 (3.59–3.65)	5.02 (4.97–5.07)	5.28 (5.24–5.33)	5.04 (5.00–5.09)	5.53 (5.48–5.58)	5.91 (5.85–5.96)
		<3.59–3.65>	<4.97–5.07>	<5.24–5.33>	<5.00–. 5.09>	<5.48–5.58>	<5.85–5.96>
Anti–reflux formulae	4						
Cu		0.23 (0.18–0.26)	0.32 (0.25–0.36)	0.34 (0.27–0.38)	0.33 (0.25–0.36)	0.36 (0.28–0.40)	0.38 (0.30–0.42)
		<**0.14**–0.28>	<**0.19**–0.38>	<0.20–040>	<**0.19**–0.38>	<0.21–0.42>	<0.23–0.45>
Zn		3.50 (3.41–3.66)	4.85 (4.78–4.98)	5.11 (5.03–5.24)	4.87 (4.80–5.00)	5.34 (5.26–5.48)	5.71 (5.62–5.85)
		<3.41–3.66>	<4.73–5.07>	<4.98–5.34>	<4.75–5.09>	<5.21–5.58>	<5.57–5.97>
Comfort–formulae	6						
Cu		0.25 (0.23–0.28)	0.35 (0.32–0.39)	0.37 (0.34–0.41)	0.35 (0.33–0.39)	0.39 (0.36–0.43)	0.41 (0.38–0.46)
		<0.20–0.31>	<0.27–0.43>	<0.29–0.45>	<0.27–0.43>	<0.30–0.47>	<0.32–0.50>
Zn		3.42 (2.80–3.82)	4.74 (3.88–5.30)	4.99 (4.08–5.58)	4.76 (3.89–5.32)	5.22 (4.27–5.83)	5.57 (4.56–6.23)
		<2.69–3.93>	<3.73–5.45>	<3.92–5.74>	<3.74–5.47>	<4.10–6.00>	<4.38–6.41>
Lactose–free formulae	2						
Cu		0.27 <0.24–0.29>	0.37 <0.34–0.40>	0.39 <0.35–0.43>	0.37 <0.34–0.41>	0.41 <0.37–0.44>	0.44 <0.39–0.48>
Zn		3.26 <2.6–3.88>	4.52 <3.65–5.38>	4.75 <3.84–5.67>	4.54 <3.66–5.41>	4.97 <4.02–5.93>	5.31 <4.29–6.33>
Preterm formulae	4 (3)						
Cu		0.31 (0.29–0.47)	0.43 (0.40–0.66)	0.46 (0.42–0.69)	0.43 (0.40–0.66)	0.48 (0.44–0.72)	0.51 (0.47–0.77)
		<0.29–0.47>	<0.40–0.66>	<0.42–0.69>	<0.40–0.66>	<0.44–0.72>	<0.47–0.77>
Zn		4.65 (3.96–4.65)	6.45 (5.49–6.45)	6.78 (5.77–6.79)	6.47 (5.51–6.48)	7.09 (6.04–7.10)	7.58 (6.45–7.59)
		<3.96–4.65>	<5.49–6.45>	<5.77–6.79>	<5.51–6.48>	<6.04–7.10>	<6.45–7.59>
Soya–based formula for infants	1						
Cu		0.21	0.28	0.30	0.29	0.31	0.33
Zn		3.99	5.53	5.82	5.55	6.09	6.50
Medium chain triglycerides formulae	3 (2)						
Cu		0.22	0.31	0.32	0.31)	0.34	0.36
		<0.22–0.22>	<0.30–0.31>	<0.32–0.33>	<0.30–0.31>	<0.33–0.34>	<0.35–0.36>
Zn		2.86	3.97	4.17	3.98	4.36	4.66
		<2.81–2.91>	<3.89–4.04>	<4.10–4.25>	<3.91–4.05>	<4.29–4.44>	<4.58–4.75>
Phenylalanine–free formulae (complete)	3						
Cu		0.27	0.38	0.40	0.38	0.42	0.45
		<0.23–0.68>	<0.32–0.95>	<0.34–0.99>	<0.32–0.95>	<0.35–**1.04**>	<0.38–**1.11**>
Zn		4.83	6.70	7.05	6.73	7.38	7.88
		<2.45–6.90>	<3.39–**9.57**>	<3.57–**10.07**>	<3.40–**9.61**>	<3.73–**10.53**>	<3.99–**11.25**>

* For the same formula from the different batch, the average value was used.

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
