# Peer review of "Copper and Zinc Content in Infant Milk Formulae Available on the Polish Market and Contribution to Dietary Intake"

_nutrients, 2021, doi:10.3390/nu13082542_

Round 1

Reviewer 1 Report

This study aimed to measure the concentrations of Zn and Cu and the Cu/Zn ratio in different types of infant formulae, compare the measurements with the manufacturer's declarations and estimate the daily intake of these trace elements.

Some aspects should be addressed and need clarification.

Major queries:

  • Zn and Cu concentrations in infant formulae have been already measured and described (Hambidge 1989, Rodríguez 2000, Khaghani, 2010) and it should be stated which relevant scientific information is added in this study.
  • Wherever it is stated that formulae were “tested” it should be replaced with “analyzed”, since no test has been essayed in this study.
  • At least one reference should support the method used for sample preparation and measurement of Zn and Cu concentrations.
  • To estimate the daily intake of Zn and Cu, which volume intake of formulae was assumed?
  • The statement (lines 229-231) “The main limitation of this study is its cross-sectional nature, which makes it impossible to assess the absorption of micronutrients…” is incorrect, since not having measured the serum concentration of these trace elements made impossible assessing their absorption and not the cross-sectional nature of the study.

Minor queries

  • The reference to zinc and copper in the manuscript should follow the same criterion. After explaining the symbol, the names should not be used in extent. For instance, in the same paragraph it is inappropriately stated “copper” in lower case (line 115), “Copper” in upper case (line 117), and “Cu” (line 118).
  • Line 22: I suggest replacing “incorrect” with “inappropriate”
  • Lines 31-32: There are non-medical conditions that may hinder breastfeeding. Therefore, I suggest replacing “medical conditions” with “constraints”
  • The statement in lines 47-49 should be supported by a reference
  • Lines 91-92: would the authors want to state “All analyses were performed in triplicate” instead of “All analyses were performed in three replications”?
  • Lines 111-112: the statement “The concentration of copper and zinc in 103 formulae for infants was determined by analysis on an atomic absorption spectrometer” should be moved to Materials and Methods section.
  • Line 139 and 192: I suggest replacing “permissible” with “recommended”
  • After explaining in extent “adequate intake (AI)” (line 145), only the abbreviation should be used and not in extent (line 153).
  • Line 186: I suggest replacing “available” with “bioavailable”
  • Line 207: replace semicolon (;) with colon (:)
  • Line 212: typo error: Zu should be replaced with Zn
  • Line 220: typo error: Ehrenkrantz should be replaced with Ehrenkranz
  • Lines 220-221: The report by Ehrenkranz et al. is correctly supported with reference 22, but reference 23 is not related with Ehrenkranz et al. report
  • Line 237: I suggest replacing “did not commit adulteration” with “provided accurate declarations of values”

References

  • Hambidge KM, Krebs NF. Upper limits of zinc, copper and manganese in infant formulas. J Nutr. 1989 Dec;119(12 Suppl):1861-4.
  • Khaghani S, Ezzatpanah H, Mazhari N, Givianrad MH, Mirmiranpour H, Sadrabadi FS. Zinc and copper concentrations in human milk and infant formulas. Iran J Pediatr. 2010 Mar;20(1):53-7.
  • Rodríguez Rodríguez EM, Sanz Alaejos M, Díaz Romero C. Concentrations of iron, copper and zinc in human milk and powdered infant formula. Int J Food Sci Nutr. 2000 Sep;51(5):373-80.

Author Response

Firstly, we would like to express our profound thanks to the Reviewer for devoting time to reviewing our manuscript, the corrections and suggestions. We have carried out a major revision of the manuscript, and we believe the paper has improved significantly.

The Reviewer's comment: Zn and Cu concentrations in infant formulae have been already measured and described (Hambidge 1989, Rodríguez 2000, Khaghani, 2010) and it should be stated which relevant scientific information is added in this study.

The authors' answer: The composition of infant milk formulae has been changing, suggesting the need for repeated analyses. In our study, we analyzed all samples of powdered milk formulae for infants available on the Polish market, including new formulae. Additionally, we paid attention to the Cu/Zn ratio.

The Reviewer's comment: Wherever it is stated that formulae were “tested” it should be replaced with “analyzed”, since no test has been essayed in this study.

The authors' answer: According to the Reviewer's suggestion, the changes have been made in the manuscript.

The Reviewer's comment: At least one reference should support the method used for sample preparation and measurement of Zn and Cu concentrations.

The authors' answer: The determination was made based on the method described by Yaman et al. [16]. We added this information to the manuscript.

  1. Yaman, M.; Çokol, N. Determination of Trace Elements in Human Milk, Cow’s Milk, and Baby Foods by Flame AAS Using Wet Ashing and Microwave Oven Sample Digestion Procedures. Atomic Spectroscopy 2004, 25, 185–190.

The Reviewer's comment: To estimate the daily intake of Zn and Cu, which volume intake of formulae was assumed?

The authors' answer: The estimated daily intake was calculated from the average energy requirements for the 0-6 months aged infants, declared by EFSA NDA Panel, with using the information on the labels about energy content and amount of powder to need to prepare 100 ml formula [4]. We added this information to the manuscript.

  1. Scientific Opinion on the Essential Composition of Infant and Follow-on Formulae. EFSA Journal, doi:10.2903/j.efsa.2014.3760.

The Reviewer's comment: The statement (lines 229-231) “The main limitation of this study is its cross-sectional nature, which makes it impossible to assess the absorption of micronutrients…” is incorrect, since not having measured the serum concentration of these trace elements made impossible assessing their absorption and not the cross-sectional nature of the study.

The authors' answer: According to the Reviewer's suggestion, the changes have been made in the manuscript: Since serum microelement concentrations were not measured, we could not assess Zn and Cu absorption.

The Reviewer's comment: The reference to zinc and copper in the manuscript should follow the same criterion. After explaining the symbol, the names should not be used in extent. For instance, in the same paragraph it is inappropriately stated “copper” in lower case (line 115), “Copper” in upper case (line 117), and “Cu” (line 118).

The authors' answer: According to the Reviewer's suggestion, the manuscript has been revised. 

The Reviewer's comment: Line 22: I suggest replacing “incorrect” with “inappropriate”

The authors' answer: According to the Reviewer's suggestion, the manuscript has been revised.

The Reviewer's comment: Lines 31-32: There are non-medical conditions that may hinder breastfeeding. Therefore, I suggest replacing “medical conditions” with “constraints”

The authors' answer: According to the Reviewer's suggestion, the changes have been made in the manuscript.

The Reviewer's comment: The statement in lines 47-49 should be supported by a reference
The authors' answer: According to the Reviewer's suggestion, we added the reference.

  1. Ljung, K.; Palm, B.; Grandér, M.; Vahter, M. High Concentrations of Essential and Toxic Elements in Infant Formula and Infant Foods – A Matter of Concern. Food Chemistry 2011, 127, 943–951, doi:10.1016/j.foodchem.2011.01.062.

The Reviewer's comment: Lines 91-92: would the authors want to state “All analyses were performed in triplicate” instead of “All analyses were performed in three replications”?

The authors' answer: According to the Reviewer's suggestion, the linguistic inaccuracy has been corrected.

The Reviewer's comment: Lines 111-112: the statement “The concentration of copper and zinc in 103 formulae for infants was determined by analysis on an atomic absorption spectrometer” should be moved to Materials and Methods section.

The authors' answer: We have been moved this piece of information to the Material and Methods section - lines 88-89.

The Reviewer's comment: Line 139 and 192: I suggest replacing “permissible” with “recommended”

The authors' answer: According to the Reviewer's suggestion, the changes have been made in the manuscript.

The Reviewer's comment: After explaining in extent “adequate intake (AI)” (line 145), only the abbreviation should be used and not in extent (line 153).

The authors' answer: According to the Reviewer's suggestion, the changes have been made in the manuscript.

The Reviewer's comment: Line 186: I suggest replacing “available” with “bioavailable”

The authors' answer: According to the Reviewer's suggestion, the changes have been made in the manuscript.

The Reviewer's comment: Line 207: replace semicolon (;) with colon (:)

The authors' answer: We have corrected it.

The Reviewer's comment: Line 212: typo error: Zu should be replaced with Zn

The authors' answer: We have corrected it.

The Reviewer's comment: Line 220: typo error: Ehrenkrantz should be replaced with Ehrenkranz

The authors' answer: According to the Reviewer's suggestion, the changes have been made in the manuscript.

The Reviewer's comment: Lines 220-221: The report by Ehrenkranz et al. is correctly supported with reference 22, but reference 23 is not related with Ehrenkranz et al. report

The authors' answer: It was an editorial mistake. We have been removed this reference.

The Reviewer's comment: Line 237: I suggest replacing “did not commit adulteration” with “provided accurate declarations of values”

The authors' answer: According to the Reviewer's suggestion, the changes have been made in the manuscript.

Reviewer 2 Report

The authors examined concentrations of copper and zinc in infant formulas that were available on the Polish market in 2019 and 2020. The manuscript is clearly written.

  1. What are Estimated Average Requirement (EAR) and Recommended Daily Allowance (RDA) for copper and zinc, respectively?
  2. What are the concentrations of copper and zinc in human milk in different lactation stages? Is the ratio of copper to zinc in human milk is 1:8 throughout lactation?
  3. What is the relationship between the estimated DI and AI of zinc?
  4. Lines 48-49, what are the adverse health outcomes?
  5. Lines 50-51, does the authors mean that deficiency of copper may impair absorption of zinc?
  6. Lines 51-52, why zinc in formulas is less available than that in breast milk?
  7. Line 61, what is the dose of copper and how long does the exposure of copper cause liver damage?
  8. Lines 68-93, based on Materials and sample preparation, each sample (can or bag) were analyzed in three replications. The replications only provide information of assay variation. Did the authors evaluate variation between the same samples in the same batch? Lines 163 and 166, for the formula from different batches with high variation, such as Cu concentration in C18 and C18’, more samples need to be analyzed to draw the conclusion.
  9. In the materials and method section, the information of reagents and equipment used in the study is incomplete.
  10. Line 123, what is the standard for the author to judge whether the data are satisfactory?
  11. Line 133, there is a typo in the title of Figure 2, Cuà
  12. The meaning of the * symbol needs to be included in the legend of Figure 1 &2.
  13. Table 1, 2 &4, should be .
  14. Lines 140-144, why do the four formulas contain lower levels of copper? Why the zinc concentrations of all tested formulas are between recommended minimum and maximum levels, but not the copper concentrations in all these infant formulas?
  15. Lines 153-155, why the zinc concentrations in all tested formulas are higher that the adequate intake?

Author Response

Firstly, we would like to express our profound thanks to the Reviewer for devoting time to reviewing our manuscript, the corrections and suggestions. We have carried out a major revision of the manuscript, and we believe the paper has improved significantly.

  1. The Reviewer's comment: What are Estimated Average Requirement (EAR) and Recommended Daily Allowance (RDA) for copper and zinc, respectively?

The authors' answer: Estimated Average Requirement (EAR) and Recommended Daily Allowance (RDA) for copper and zinc are not available for infants in this period. Instead of it EFSA and ASPEN panel recommends using adequate intake (AI).

Vanek, V.W.; Borum, P.; Buchman, A.; Fessler, T.A.; Howard, L.; Jeejeebhoy, K.; Kochevar, M.; Shenkin, A.; Valentine, C.J.; Novel Nutrient Task Force, Parenteral Multi-Vitamin and Multi–Trace Element Working Group; et al. A.S.P.E.N. Position Paper: Recommendations for Changes in Commercially Available Parenteral Multivitamin and Multi–Trace Element Products. Nutr Clin Pract 2012, 27, 440–491, doi:10.1177/0884533612446706.

Scientific Opinion on the Essential Composition of Infant and Follow-on Formulae. EFSA Journal, doi:10.2903/j.efsa.2014.3760.

  1. The Reviewer's comment: What are the concentrations of copper and zinc in human milk in different lactation stages

The authors' answer:. The copper content in colostrum amounts to app. 520 μg/L; it decreases to 350-450 μg/L at 1st month of lactation, and 200 μg/L at 6th month of lactation.  Zinc content is also variable and amounts to 4 mg/L in colostrum, 1.75 mg/L at 1st month of lactation, and 0.7 mg/L at 6th month of lactation.

Dietary Reference Intakes for Vitamin A, Vitamin K, Arsenic, Boron, Chromium, Copper, Iodine, Iron, Manganese, Molybdenum, Nickel, Silicon, Vanadium, and Zinc; National Academies Press: Washington, D.C., 2001; p. 10026; ISBN 978-0-309-07279-3. https://doi.org/10.17226/10026.

  1. The Reviewer's comment: Is the ratio of copper to zinc in human milk is 1:8 throughout lactation?

The authors' answer: The Cu/Zn ratio changes at different stages of lactation. It is the highest in colostrum (1:8); it decreases to 1:5 in the 1st month of lactation and 1:3.5 in the 6th month of lactation.

  1. The Reviewer's comment: What is the relationship between the estimated DI and AI of zinc?

The authors' answer: The calculation of AI is based on experimental estimates of nutrient intake by a group (or groups) of healthy people and theoretically, the daily intake should be equal to the AI. However, our study showed that the adequate intake does not coincide with the estimated DI (based on the expected consumption of infant milk formula) in all cases.

  1. The Reviewer's comment: Lines 48-49, what are the adverse health outcomes?

The authors' answer: This sentence has been elaborated on: A concentration of elements in formulae for infants, which are either too low or too high, can lead to adverse health outcomes (such as impaired growth, increased morbidity, impaired cognitive development and hyperactivity) [7].

  1. Ljung, K.; Palm, B.; Grandér, M.; Vahter, M. High Concentrations of Essential and Toxic Elements in Infant Formula and Infant Foods – A Matter of Concern. Food Chemistry 2011, 127, 943–951, doi:10.1016/j.foodchem.2011.01.062.

  1. The Reviewer's comment: Lines 50-51, does the authors mean that deficiency of copper may impair absorption of zinc?

The authors' answer: The inaccuracy has been corrected: Excessive Zn intake can reduce intestinal copper absorption.

  1. The Reviewer's comment: Lines 51-52, why zinc in formulas is less available than that in breast milk?

The authors' answer: An important factor in low zinc bioavailability from the infant milk formulae is lower whey-to-casein ratio than in human milk. Zinc that is bound to casein is less bioavailable compared with zinc bound to whey proteins.

Ackland, M.L.; Michalczyk, A.A. Zinc and Infant Nutrition. Archives of Biochemistry and Biophysics 2016, 611, 51–57, doi:10.1016/j.abb.2016.06.011.

  1. The Reviewer's comment: Line 61, what is the dose of copper and how long does the exposure of copper cause liver damage?

The authors' answer: We corrected this sentence in the manuscript.: Toxic effects associated with increased chronic exposure to Cu cause acute gastrointestinal symptoms: abdominal pain, vomiting, diarrhea.

The lowest observed adverse effect level (LOAEL) was determined at 6 mg Cu/L, and the no observed adverse effect level (NOAEL) for gastrointestinal effects and nausea was 4 mg Cu/L. However, oral exposure of copper does not seem to result in liver damage.

Taylor, A.A.; Tsuji, J.S.; Garry, M.R.; McArdle, M.E.; Goodfellow, W.L.; Adams, W.J.; Menzie, C.A. Critical Review of Exposure and Effects: Implications for Setting Regulatory Health Criteria for Ingested Copper. Environmental Management 2020, 65, 131–159, doi:10.1007/s00267-019-01234-y.

  1. The Reviewer's comment: Lines 68-93, based on Materials and sample preparation, each sample (can or bag) were analyzed in three replications. The replications only provide information of assay variation. Did the authors evaluate variation between the same samples in the same batch? Lines 163 and 166, for the formula from different batches with high variation, such as Cu concentration in C18 and C18’, more samples need to be analyzed to draw the conclusion.

The authors' answer: Indeed, you are right (variation was changed on differences). We also added these sentences to chapter of limitation of the study:

In our study, we analyzed 90 samples from all formulae available on the Polish market; in cases of 13 formulae the measurement was repeated on samples derived from separate batches. Unfortunately, due to the lack of sufficient funds, we were not able to analyze more samples from different batches of the same formula.

  1. The Reviewer's comment: In the materials and method section, the information of reagents and equipment used in the study is incomplete.

The authors' answer: According to the Reviewer's suggestion, the changes have been made in the manuscript.

  1. The Reviewer's comment: Line 123, what is the standard for the author to judge whether the data are satisfactory?

The authors' answer: According to the recent recommendations, a deviation of 19% was adopted.

Konings, E.J.M.; Roux, A.; Reungoat, A.; Nicod, N.; Campos-Giménez, E.; Ameye, L.; Bucheli, P.; Alloncle, S.; Dey, J.; Daix, G.; et al. Challenge to Evaluate Regulatory Compliance for Nutrients in Infant Formulas with Current State-of-the-Art Analytical Reference Methods. Food Control 2021, 119, 107423, doi:https://doi.org/10.1016/j.foodcont.2020.107423.

  1. The Reviewer's comment: Line 133, there is a typo in the title of Figure 2, Cuà

The authors' answer: We have corrected it.

  1. The Reviewer's comment: The meaning of the * symbol needs to be included in the legend of Figure 1 &2.

The authors' answer: According to the Reviewer's suggestion, the changes have been made in the manuscript (* randomly assigned formulae symbols from two other batches).

  1. The Reviewer's comment: Table 1, 2 &4, should be .

The authors' answer: According to the Reviewer's suggestion, the changes have been made in the manuscript.

  1. The Reviewer's comment: Lines 140-144, why do the four formulas contain lower levels of copper? Why the zinc concentrations of all tested formulas are between recommended minimum and maximum levels, but not the copper concentrations in all these infant formulas?

The authors' answer: It is probably related to the differences between the recommended minimum levels of copper in formulae for special medical purposes for infants (0.06 mg/100kcal) and “regular” infant formulae (0.035 mg/kcal) and follow-on formulae (0.035 mg/kcal). All analyzed 4 formulae for special medical purposes met the limits on copper in infant and follow-on formulae. The underlying reason might be that some of the formulae for special medical purposes for infants are intended to be used both by infants aged 0-6 and 6-12 months as declared by the producing company. On the other hand, no differences in minimum level for zinc between special medical purposes and infant and follow-on formulae exist. Besides, the recommended ranges are wider.

  1. The Reviewer's comment: Lines 153-155, why the zinc concentrations in all tested formulas are higher that the adequate intake?

The authors' answer: The zinc concentrations are not higher than the recommended levels. However, the estimated intake based on age-related recommendations indeed is higher This could be at least in part explained by the lower zinc bioavailability from infant formulae than from human milk.

Round 2

Reviewer 1 Report

The revised version of the manuscript is improved, but some minor issues still need to be addressed:

  • “In all cases” stated in line 185 is confusing and incoherent with the statement “In most cases” in lines 25 and 253. Please be accurate.
  • The objective stated in the abstract should be coherent with that stated in the text of the manuscript (lines 66-69), that is, it should be specified that Cu and Zn concentration were analyzed in formulae for infants available in the Polish market.
  • Line 13: please correct the typo error “infants” instead of “infantas”
  • Lines 20 and 125: I would suggest replacing “producer” with “manufacturer”
  • Line 220: I would suggest replacing “producing company” with “manufacturer”
  • Lines 64-65: instead of “…gastrointestinal symptoms: abdominal pain, vomiting, diarrhea” I would suggest “…gastrointestinal symptoms, such as abdominal pain, vomiting, and diarrhea”
  • Line 89: I suggest eliminating “by analysis” and “made”
  • Line 116: I suggest eliminating “with”
  • Line 207: I suggest stating in plural “requirements” and eliminating “micronutrients”
  • The statement in lines 227-228, started with “Zinc that is bound to casein…” should be supported with a reference

Author Response

Dear Reviewer, we appreciate all your insightful comments. Thank you for your suggestions.

The Reviewer's comment: The revised version of the manuscript is improved, but some minor issues still need to be addressed:

  • “In all cases” stated in line 185 is confusing and incoherent with the statement “In most cases” in lines 25 and 253. Please be accurate.
  • The objective stated in the abstract should be coherent with that stated in the text of the manuscript (lines 66-69), that is, it should be specified that Cu and Zn concentration were analyzed in formulae for infants available in the Polish market.
  • Line 13: please correct the typo error “infants” instead of “infantas”
  • Lines 20 and 125: I would suggest replacing “producer” with “manufacturer”
  • Line 220: I would suggest replacing “producing company” with “manufacturer”
  • Lines 64-65: instead of “…gastrointestinal symptoms: abdominal pain, vomiting, diarrhea” I would suggest “…gastrointestinal symptoms, such as abdominal pain, vomiting, and diarrhea”
  • Line 89: I suggest eliminating “by analysis” and “made”
  • Line 116: I suggest eliminating “with”
  • Line 207: I suggest stating in plural “requirements” and eliminating “micronutrients”
  • The statement in lines 227-228, started with “Zinc that is bound to casein…” should be supported with a reference

The authors' answer: All changes have been made in the manuscript and highlighted in red.

References

[1] Ackland, M.L.; Michalczyk, A.A. Zinc and Infant Nutrition. Arch Biochem Biophys 2016, 611, 51–57, doi:10.1016/j.abb.2016.06.011.
